# Level-set Parameters:
# Novel Data for 3D Shape Analysis

## Abstract

3D shape analysis has been largely focused on traditional 3D data of point clouds and meshes, but the discrete nature of these data makes the analysis susceptible to variations in input resolutions. Recent development of *neural fields* brings in level-set parameters from signed distance functions as a novel, continuous, and numerical representation of 3D shapes, where the shape surfaces are defined as zero-level-sets of those functions. This motivates us to extend shape analysis from the traditional 3D data to these novel parameter data. Since the level-set parameters are not Euclidean like point clouds, we establish correlations across different shapes by formulating them as a pseudo-normal distribution, and learn the distribution prior from the respective dataset. To further explore the level-set parameters with shape transformations, we propose to condition a subset of these parameters on rotations and translations, and generate them with a hyper-network. We demonstrate the potential of the novel continuous representation in pose-related shape analysis through applications to shape classification, retrieval under arbitrary poses, and 6D object pose estimation. Code and data in this research are anonymously provided at this github link.

## 1 Introduction

3D surfaces are traditionally represented as point clouds or meshes on digital devices for visualization and geometry processing. This convention results in the current predominance of those data in 3D shape analysis, though the discrete nature of point clouds and polygon meshes can make the analysis approaches susceptible to variations in data resolutions (Qi et al., 2017a; Wang et al., 2019d; Li et al., 2018; Lei et al., 2020; Hanocka et al., 2019; Hu et al., 2022). The recent advancements in neural fields enable manifold surfaces to be continuously represented by the zero-level-set of their signed distance functions (SDFs) (Park et al., 2019a; Sitzmann et al., 2020b; Xie et al., 2022). Specifically, SDFs compute a scalar field by mapping each coordinate $\mathbf{x} \in \mathbb{R}^3$ to a scalar $v \in \mathbb{R}$ using a deep neural network, where the scalar $v$ indicates the signed distance of a point to its closest point on the surface boundary. Let $\boldsymbol{\theta}$ be the optimized parameters of the neural network. The shape surface can be numerically represented as $\boldsymbol{\theta}$, which we refer to as the *level-set parameters*. A typical example of 3D surface representation with level-set parameters could be $[n_x, n_y, n_z, d]$ for a 3D plane of $\mathbf{n}^\mathsf{T}\mathbf{x} + d = 0$, where $\mathbf{n} = [n_x, n_y, n_z]^\mathsf{T}$ and $\|\mathbf{n}\| = 1$. The level-set parameters bring in novel 3D data for continuous shape analysis.

To initiate the novel shape analysis, we first have to construct the level-set parameters for each shape *independently* and with good accuracy in the dataset of interest. This differs from most methods in the SDF research domain, which are reconstruction-oriented and use *a shared decoder* together with various latent codes to improve the surface quality (Park et al., 2019a; Mescheder et al., 2019; Erler et al., 2020; Chen & Zhang, 2019; Peng et al., 2020; Chibane et al., 2020). However, the level-set parameters do not conform to Euclidean or metric geometry, unlike 3D point clouds. This presents a critical challenge for establishing good correlations in the parameter data of different shapes. Our solution is to formulate those parameters in high dimension as a pseudo-normal distribution with expectation $\boldsymbol{\mu}$ and identity covariance matrix $\mathbf{I}$, that is,

$$\boldsymbol{\theta} = \boldsymbol{\mu} + \Delta\boldsymbol{\theta}. \tag{1}$$

We associate $\Delta\boldsymbol{\theta}$ with each individual shape, while the parameter $\boldsymbol{\mu}$ is shared by all shapes to (i) align different shapes in the parameter space, and (ii) initialize the SDF networks of individual

shapes. Previous works train *per-category* SDF initializations using all training shapes (Sitzmann et al., 2020a) or a particular shape (Erkoç et al., 2023) in the category, which are either computation-intensive or dependent on the specific shape chosen. In contrast, we seek an initialization that generalizes to all categories of the dataset. Therefore, we take the trade-off of Sitzmann et al. (2020a) and Erkoç et al. (2023) to use a few shapes from *each category* to learn $\mu$. Other than learning an initialization, there are also methods employing identical random settings to initialize (Luigi et al., 2023), which we empirically show yield insufficient correlations among shapes.

Level-set parameters have rarely been considered as a data modality in shape analysis. Luigi et al. (2023) had to rely on the traditional data (*i.e.*, point cloud and meshes) to extract shape semantics from level-set parameters using an encoder-decoder, due to their suboptimal SDF initializations. Erkoç et al. (2023) utilized a small SDF network for tractable complexity in shape diffusion, whereas a small SDF network can undermine the representation quality of level-set parameters for complex shapes. More importantly, both methods are limited to shape analysis in the reference poses, ignoring the important shape transformations such as rotation and translation. In contrast, we explore the level-set parameters with transformations and extend the shape analysis to be pose-related.

Usually, transformations of shapes represented by level-set parameters only affect a typical subset of the parameters (*e.g.*, those in the first layer of SDF). We propose to condition those subset parameters on rotations and translations and generate them with a hypernetwork to facilitate the analysis. The inherent separation of pose-dependent and pose-independent subsets in level-set parameters enables classification of continuous shapes in arbitrary poses to be simple and outperform equivariant neural networks for point clouds (Deng et al., 2021; Chen & Cong, 2022). Our work focuses on demonstrating the viability of level-set parameters as an independent data modality for continuous shape analysis. This contrasts with existing approaches that leverage neural fields to learn invariant features from discrete data for semantic analysis (Kwon et al., 2023).

To acquire experimental data, we construct level-set parameters for shapes in the ShapeNet (Chang et al., 2015) and Manifold40 (Hu et al., 2022; Wu et al., 2015) datasets. We demonstrate the potential of the proposed data through applications in shape classification of arbitrary poses, shape retrieval, and 6D object pose estimation. In pose estimation, we consider the problem of estimating shape poses from their partial point cloud observations, given that the level-set parameters are provided. This is similar to a partial-to-whole registration task (Dang et al., 2022). The main contributions of this work are summarized as below:

- We introduce level-set parameters as a novel data modality for 3D shapes, and demonstrate their potential with pose-related shape analysis. A new hypernetwork is contributed to transform the shapes in the level-set parameter space to facilitate the analysis.
- We present an encoder that is able to accept the high-dimensional level-set parameters as inputs and extract the shape semantics in arbitrary poses for classification and retrieval.
- We propose a correspondence-free registration approach that estimates the 6D object poses from their partial point cloud observations based on the SDF reconstruction loss.
- We opensource our code and level-set parameter data on github.

## 2 RELATED WORK

### 2.1 SHAPE ANALYSIS WITH TRADITIONAL 3D DATA

**Semantic Analysis.** The traditional 3D data, point clouds and polygon meshes, represent shape surfaces discretely in Euclidean space. Point clouds, as orderless collections of points, result in early-stage MLP-based neural networks (Qi et al., 2017a; Klokov & Lempitsky, 2017; Qi et al., 2017b) to introduce permutation-invariant operations for semantic learning from such data. Graph convolutional networks (Lei et al., 2020; Wang et al., 2019c; Wu et al., 2019; Thomas et al., 2019) enable point clouds to be processed with convolutional operations which handle spatial hierarchies of data better than MLPs. Transformer-based networks (Zhao et al., 2021; Guo et al., 2021; Wu et al., 2022) treat each point cloud as a sequence of 3D points, offering another approach. Point clouds can also be voxelized into regular grids and processed by 3D convolutional neural networks (Wu et al., 2015; Maturana & Scherer, 2015; Riegler et al., 2017; Graham et al., 2018). Polygon meshes, which are point clouds with edge connections on the shape surface, provide more information about the shape geometry. Different approaches have been proposed to learn shape semantics from meshes (Hanocka et al., 2019; Hu et al., 2022; Smirnov & Solomon, 2021; Lei et al., 2023).

Furthermore, a variety of equivariant networks have been presented to incorporate rotation and translation equivariance for the method to maintain its effectiveness when the discrete data are transformed (Deng et al., 2021; Chen & Cong, 2022; Esteves et al., 2018; Cohen et al., 2018; Thomas et al., 2018; Poulenard & Guibas, 2021). In contrast, we study shape semantics from their continuous representations using level-set parameters, requiring no equivariant modules for transformations.

**Geometric Analysis.** 6D object pose estimation is crucial for various applications such as robotics grasping (Tremblay et al., 2018), augmented reality (Marchand et al., 2015), and autonomous driving (Geiger et al., 2012). It requires accurately determining the rotation and translation of an object relative to a reference frame. Despite the predominance of estimating object poses from RGB(D) data (Wang et al., 2019a;b; Peng et al., 2019; Park et al., 2019b), this task can alternatively be defined as a registration problem based on point cloud data. It involves registering the partial point cloud of an object (observation) to its full point cloud (reference) (Dang et al., 2022; Jiang et al., 2023). We are interested in the potential of level-set parameters in the geometric analysis of 3D shapes and therefore represent the reference objects using level-set parameters instead of full point clouds. Leveraging the SDF reconstruction loss, we propose a registration method that requires no training data (Zeng et al., 2017), correspondences (Choy et al., 2019; Wang & Solomon, 2019; Huang et al., 2021; Ao et al., 2023) or global shape features (Huang et al., 2020; Aoki et al., 2019). We compare it with other optimization-based registration algorithms that also require no training data, including ICP (Besl & McKay, 1992), FGR (Zhou et al., 2016), and TEASER(++) (Yang et al., 2020). Go-ICP (Yang et al., 2015) is excluded here due to its high time complexity.

## 2.2 NEURAL FIELDS FOR 3D RECONSTRUCTION

Neural fields utilize coordinate-based neural networks to compute signed distance fields (Park et al., 2019a; Sitzmann et al., 2020b) or occupancy fields (Mescheder et al., 2019) for 3D reconstructions. Many methods in this domain employ a shared decoder to reconstruct 3D shapes across an entire dataset or a specific category using different shape latent codes (Park et al., 2019a; Mescheder et al., 2019; Erler et al., 2020; Chen & Zhang, 2019; Atzmon & Lipman, 2020; Gropp et al., 2020). The modelling function can be denoted as $f_{\boldsymbol{\theta}} : \mathbb{R}^3 \times \mathbb{R}^m \rightarrow \mathbb{R}$, with $\mathbf{z} \in \mathbb{R}^m$ being the shape latent code, which is learned by an encoder from various inputs (Mescheder et al., 2019), or randomly initialized and optimized as in DeepSDF (Park et al., 2019a). Some works extend the concept of latent codes from per-shape to per-point for improved surface quality (Peng et al., 2020; Chibane et al., 2020). In contrast, SIREN (Sitzmann et al., 2020b) applies an SDF network to instance-level surface reconstruction using periodic activations, which does not involve latent codes. It employs an unsupervised loss function, eliminating the need for ground-truth SDF values as in DeepSDF. We note that neural radiance fields (NeRFs) (Martin-Brualla et al., 2021; Wang et al., 2021; Yu et al., 2022; Yariv et al., 2021) reconstruct 3D surfaces with entangled neural parameters for surface geometry and photometry, where the latter is view-dependent. We therefore focus on surface geometry and utilize the level-set parameters from SDFs in our study. Our SDF network is adapted from the common 8-layer MLP utilized by others (Park et al., 2019a; Atzmon & Lipman, 2020; Gropp et al., 2020).

## 2.3 LEVEL-SET PARAMETERS AS 3D DATA

Few works have studied the level-set parameters as an alternative data modality for 3D research, and each has utilized a different SDF network. Luigi et al. (2023) initialized the SDF network of SIREN (Sitzmann et al., 2020b) with identical random settings and trained the parameters independently for each shape, resulting in insufficient shape correlations in the parameter space. To extract shape semantics from the level-set parameters, theyo utilized an encoder-decoder architecture with additional supervision from traditional data. In contrast, our proposed method constructs the parameter data with improved shape correlations, enabling the learning of shape semantics using an encoder without relying on traditional data. We note that the periodic activation functions of SIREN cause undesired shape artifacts in empty spaces compared to ReLU activations (Ben-Shabat et al., 2022). Erkoç et al. (2023) utilized a much smaller SDF network with ReLU activations for continuous shape generation. They initialized the network using overfitted parameters of a particular shape and trained the parameters of each shape within the same category independently. Yet, small SDF networks cannot represent complex shapes with good accuracy. Dupont et al. (2022) aimed to explore the parameters of diverse neural fields as continuous data representations, but instead resorted to their modulation vectors (Mehta et al., 2021; Chan et al., 2021) for simplicity. They exploited the strategy of meta-learning (Sitzmann et al., 2020a; Finn et al., 2017; Tancik et al., 2021) to construct those modulation vectors.

The existing works are restricted to shape analysis in reference poses. We leverage the potential of level-set parameters and explore them with rotations and translations in pose-related analysis.

## 3 DATASET OF LEVEL-SET PARAMETERS

**Preliminaries.** We adopt the well-established SDF architecture from previous research (Park et al., 2019a; Atzmon & Lipman, 2020), which comprises an 8-layer MLP with a skipping concatenation at the 4th layer. We utilize 256 neurons for all interior layers other than the skipping layer which has 253 neurons due to the input concatenation. Note that shape latent codes are not required. We employ smoothed ReLU as the activation function and train the SDF network $f(\mathbf{x}; \boldsymbol{\theta})$ with the unsupervised reconstruction loss of SIREN (Sitzmann et al., 2020b), *i.e.*,

$$\mathcal{L}_{\text{SDF}} = \lambda_1 \mathcal{L}_{\text{dist}}^p + \lambda_2 \mathcal{L}_{\text{dist}}^n + \lambda_3 \mathcal{L}_{\text{eik}} + \lambda_4 \mathcal{L}_{\text{norm}}^p. \tag{2}$$

$\mathcal{L}_{\text{dist}}^p, \mathcal{L}_{\text{dist}}^n$ are the respective distances of positive and negative points to the surface. $\mathcal{L}_{\text{eik}}$ is the Eikonal loss (Gropp et al., 2020) and $\mathcal{L}_{\text{norm}}^p$ imposes normal consistency. The constants $\lambda_{1\sim4}$ balance different objectives.

The SDF network only provides level-set parameters for each shape in their reference poses. However, pose-related shape analysis requires surface transformations to be enabled in the level-set parameter space. To address this, we propose a hypernetwork that conditions a subset of the SDF parameters on rotations and translations in SE(3) in § 3.1.

### 3.1 SURFACE TRANSFORMATION

We introduce a hypernetwork $h_{\boldsymbol{\phi}}$ conditioned on rotations $\mathbf{R}$ and translations $\mathbf{t}$ to generate weights and biases for the first SDF layer. Let $m \in [256]$ be the row index, and $n \in [4]$ be the column index, where $[i] = \{1, 2, \ldots, i\}$. We index each parameter to be generated for the first SDF layer as $\theta_1^{mn}$. The trainable parameters $\boldsymbol{\phi}$ in the hypernetwork are composed of two components, including (1) the neural parameters $\boldsymbol{\eta}$ of all fully connected layers, (2) the small latent matrices $\{\mathbf{Y}^{mn} \in \mathbb{R}^{I \times J}\}$ for each $\theta_1^{mn}$ in the first SDF layer. We use $I = 2$ and $J = 8$ as dimensions of $\mathbf{Y}^{mn}$ in our experiments. The computation of each $\theta_1^{mn}$ from the hypernetwork is expressed as

$$\theta_1^{mn} = h(\mathbf{R}, \mathbf{t}; \boldsymbol{\eta}, \mathbf{Y}^{mn}). \tag{3}$$

Instead of generating the first layer parameters directly as $h(\mathbf{R}, \mathbf{t}; \boldsymbol{\eta})$, we introduce the latent matrices $\{\mathbf{Y}^{mn}\}$ to guarantee that our generated SDF parameters satisfy the geometric initializations recommended by SAL (Atzmon & Lipman, 2020). This is critical for the network convergence and good performance.

Generally, the hypernetwork calculates a compact matrix $\mathcal{B} = h(\mathbf{R}, \mathbf{t}; \boldsymbol{\eta})$ of size $256 \times 4 \times I \times J$ according to $\mathbf{R}, \mathbf{t}$, which contains the pose-dependent coefficient matrices $\mathbf{B}^{mn}$ associated with each latent matrix $\mathbf{Y}^{mn}$. Each pair of matrices $\mathbf{B}^{mn}$ and $\mathbf{Y}^{mn}$ is combined to compute a vector $\mathbf{z}^{mn}$, normalized into $\hat{\mathbf{z}}^{mn}$ to satisfy the standard normal distribution. In the last layer of $h_{\boldsymbol{\phi}}$, two branches of fully connected layers accept $\mathbf{z}^{mn}$ and $\hat{\mathbf{z}}^{mn}$, respectively, to initialize the biases and weights in $\{\theta_1^{mn}\}$ accordingly. See appendix for the details. We note that the tanh activation in the final computation of $\mathcal{B}$ (see Fig. B) helps to constrain all of its values within the range of $[-1, 1]$.

By incorporating this hypernetwork $h_{\boldsymbol{\phi}}$ into the SDF network, we obtain a transformation-enabled SDF architecture, referred to as HyperSE3-SDF. It can transform the surface in the level-set parameter space by adaptively modifying $\{\theta_1^{mn}\}$ in $\boldsymbol{\theta}$. Although we can also apply the formulas

$$\mathbf{W}' = \mathbf{W}\mathbf{R}^{-1}, \mathbf{b}' = -\mathbf{W}\mathbf{R}^{-1}\mathbf{t} + \mathbf{b}, \tag{4}$$

to the weights $\mathbf{W}$ and biases $\mathbf{b}$ in the 1st and 5th layers of the utilized SDF network for simplified surface transformations, the Euclidean nature of these computations yields transformed parameters with shape semantics that are incomparable to those produced by HyperSE3-SDF. We show this empirically with experiments. Derivations of the formulas are provided in the appendix.

### 3.2 DATASET CONSTRUCTION

As introduced in Eq. (1), we decompose the level-set parameters $\boldsymbol{\theta}$ into $\boldsymbol{\mu} + \Delta\boldsymbol{\theta}$. This is inspired by the reparameterization trick in variational autoencoders (Kingma et al., 2019). It emulates a normal distribution with expectation $\boldsymbol{\mu}$ and homogeneous standard deviation 1. We follow this decomposition to construct a dataset of level-set parameters with shape transformations in two stages.

In the first stage, we train HyperSE3-SDF with a small number of shapes to obtain the shared prior $\boldsymbol{\mu}$. The parameters in $\boldsymbol{\mu}$ are divided into (1) the pose-dependent $\{\mu_1^{mn}\}$ for the first SDF layer, and (2) the remaining $\{\mu_\ell^{mn}|\ell>1\}$. Following Eq. (3), each $\mu_1^{mn}$ is computed as

$$\mu_1^{mn} = h(\mathbf{R}, \mathbf{t}; \boldsymbol{\eta}, \mathbf{Y}^{mn}). \qquad (5)$$

We propose Algorithm 1 in the right to train the parameters $\boldsymbol{\eta}$, $\{\mathbf{Y}^{mn}\}$, and $\{\mu_\ell^{mn}|\ell>1\}$ in HyperSE3-SDF.

**Algorithm 1**

1: **for** each batch of shapes **do**
2:     Get their points and normals $\{\mathbf{P}_b, \mathbf{N}_b\}$.
3:     Sample a batch of transformations $\{\mathbf{R}_b, \mathbf{t}_b\}$.
4:     Transform the shapes as $\{\mathbf{P}_b\mathbf{R}_b+\mathbf{t}_b, \mathbf{N}_b\mathbf{R}_b\}$.
5:     Get their SDF parameters $\{\boldsymbol{\theta}_b\}$ for $\{\mathbf{R}_b, \mathbf{t}_b\}$.
6:     Compute SDF values with $\{f(\mathbf{P}_b\mathbf{R}_b+\mathbf{t}_b; \boldsymbol{\theta}_b)\}$.
7:     Calculate the SDF loss based on Eq. (7).
8:     Compute gradients and update the parameters.
9: **end for**

In Algorithm 1, $\mathbf{P}_b$ and $\mathbf{N}_b$ represent the point cloud and point normals of a shape $b$, respectively. $\mathbf{R}_b$ and $\mathbf{t}_b$ denote the random transformation applied, and $\boldsymbol{\theta}_b$ is the corresponding SDF parameters. It shares the parameters $\{\mu_\ell^{mn}|\ell>1\}$ with $\boldsymbol{\mu}$, but not $\{\mu_1^{mn}\}$. Instead, we compute each $\theta_1^{mn}$ in $\boldsymbol{\theta}_b$ by replacing the $\mathbf{Y}^{mn}$ in Eq. (3) with $\mathbf{Y}^{mn}+\Delta\mathbf{Y}^{mn}$, where $\{\Delta\mathbf{Y}^{mn}\}$ are the trainable matrices associated with the specific shape. We find this strategy necessary for network convergence. After training, we discard $\{\Delta\mathbf{Y}^{mn}\}$. The optimized parameters $\boldsymbol{\eta}$, $\{\mathbf{Y}^{mn}\}$, $\{\mu_\ell^{mn}|\ell>1\}$ will be frozen.

In the second stage, we initialize HyperSE3-SDF with the frozen parameters from stage one and train $\Delta\boldsymbol{\theta}$ for each individual shape. The trainable parameters associated with each shape include $\{\Delta\mathbf{Y}^{mn}\}$ and $\{\Delta a_\ell^{mn}|\ell>1\}$, all initialized as zeros. We compute each element in $\Delta\boldsymbol{\theta}$ as

$$\Delta\theta_\ell^{mn} = \begin{cases} \frac{1}{I\times J}\sum_{i,j}\mathbf{B}^{mn}\odot\tanh(\Delta\mathbf{Y}^{mn}), \ \ell=1; \\ \tanh(\Delta a_\ell^{mn}), \ \ell>1, \end{cases} \qquad (6)$$

with $\{\Delta\theta_1^{mn}\}$ being pose-dependenet. $\odot$ denotes the Hadamard product. The tanh function constrains $\Delta\boldsymbol{\theta}$ to be in the range $[-1, 1]$. We train the above parameters similarly based on Algorithm 1. The major difference from stage one is that all shapes in a batch become clones of the individual shape being fitted. In addition, note that $\boldsymbol{\theta} = \boldsymbol{\mu} + \Delta\boldsymbol{\theta}$ in the Algorithm in this stage.

Due to the learned initializations from stage one, we can obtain the level-set parameters of each shape with hundreds of training iterations in stage two. This facilities the acquisition of a dataset of level-set parameters with transformations, and enhances shape correlations in the parameter space.

**Why not meta-learning like MetaSDF?** MetaSDF (Sitzmann et al., 2020a) applies the meta-learning technique (Finn et al., 2017) to per-category shape reconstruction of DeepSDF. It introduces a large number of additional parameters (*i.e.*, per-parameter learning rates) to train the network, which is computation-intensive. Besides, its three gradient updates of the network during inference stage often result in unsatisfactory surface quality (Chou et al., 2022). Moreover, the loss function we employ in Eq. (7) for unsupervised surface reconstruction requires input gradients of the SDF network to be computed with backpropagation at every iteration, prohibiting the application of MetaSDF or MAML (Finn et al., 2017).

## 4 SHAPE ANALYSIS WITH LEVEL-SET PARAMETERS

### 4.1 ENCODER-BASED SEMANTIC LEARNING

We format the level-set parameters into multiple tensors for shape analysis. Specifically, the weights and biases in the first SDF layer are concatenated into a tensor of size $256\times4$. Further, we concatenate all parameters from layer 2 to 7 into a tensor of size $6\times256\times257$, while zero-padding is applied to the parameters of the skipping layer. Regarding the final SDF layer, its parameters are combined into a tensor of size $1\times257$. Thus, each shape surface is continuously represented as a tuple of three distinct tensors $(\Theta_1\in\mathbb{R}^{256\times4}, \Theta_2\in\mathbb{R}^{6\times256\times257}, \Theta_3\in\mathbb{R}^{1\times257})$. See appendix for an illustration.

The proposed semantic learning network has three branches, each processing a different component in the input tensors $(\Theta_1, \Theta_2, \Theta_3)$, as depicted in Fig. 2(a). It builds upon the BaseNet and BasePool blocks shown in Fig. 2(b). The BaseNet block comprises two fully connected layers followed by batch normalization (Ioffe & Szegedy, 2015). The first layer applies ReLU activation. For batch normalization, we *flatten* the first two dimensions of the input tensor. For example, the resulting dimensions for $\Theta_1$, $\Theta_2$, $\Theta_2^\mathsf{T}$, $\Theta_3$ will be $1024$, $1536\times257$, $1542\times256$, $257$, where $\Theta_2^\mathsf{T}$ indicates

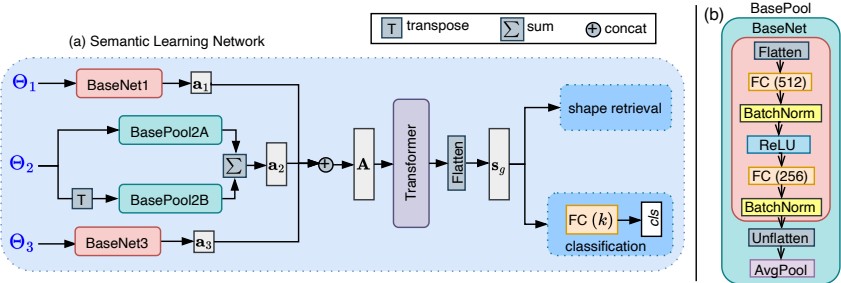

Figure 2: Encoder-based semantic learning from level-set parameters. The network in (a) processes input tensors $\Theta_1$, $\Theta_2$, and $\Theta_3$ with different branches. Their outputs are concatenated into a unified shape feature $\mathbf{A} \in \mathbb{R}^{8 \times 256}$, which is then processed by a single-layer transformer to obtain a global shape feature $\mathbf{s}_g$. We apply this global feature to shape classification and retrieval. BaseNet1 and BaseNet3 follow configurations of the BaseNet block in (b), while BasePool2A and BasePool2B use the BasePool block in (b), comprising BaseNet followed by unflattening and average pooling.

the transpose of $\Theta_2$. The BasePool block consists of BaseNet followed by unflattening and average pooling, which are required for processing the three-dimensional $\Theta_2$ and $\Theta_2^\mathsf{T}$. Specifically, we perform average pooling along the second dimension of the unflattened features.

The BaseNet1 and BaseNet3 modules compute branch features $\mathbf{a}_1 \in \mathbb{R}^{256}$ and $\mathbf{a}_3 \in \mathbb{R}^{256}$, respectively, based on $\Theta_1$ and $\Theta_3$. Meanwhile, BasePool2A and BasePool2B each compute features of identical size $6 \times 256$, which are summed to form the branch feature $\mathbf{a}_2$. These branch features are then concatenated to produce a unified surface feature $\mathbf{A} \in \mathbb{R}^{8 \times 256}$, which is subsequently processed by a single-layer transformer (Vaswani et al., 2017). We flatten the output feature to form a global shape feature $\mathbf{s}_g \in \mathbb{R}^{2048}$ and apply it to both shape classification and retrieval. The network is trained using cross-entropy loss, and the classifier comprises a single fully connected layer.

Our analysis considers level-set parameters from all layers of SDF, which forms a complete shape representation. In contrast, prior works utilized partial level-set parameters. For instance, Luigi et al. (2023) did not include the first layer parameters, while Erkoç et al. (2023) excluded the final layer parameters. Further, given our proposed decomposition of $\boldsymbol{\theta}$ and the fact that $\boldsymbol{\mu}$ is shared by all shapes, we normalize the level-set parameters with $\boldsymbol{\mu}$ and study shape semantics with $\Delta\boldsymbol{\theta}$, the instance parameters of each shape. Additionally, we go beyond reference poses and evaluate the network in semantic learning from continuous shapes under different transformation settings.

### 4.2 Registration-based 6D Pose Estimation

We train the SDF network to estimate shape poses, which entails incorporating the pose parameters $\mathbf{R}, \mathbf{t}$ into the optimizable parameters of the plain SDF network. To achieve, we can either (1) generate the 1st layer parameters with the hypernetwork, or (2) compute the pose-dependent parameters in the 1st and 5th layers based on Eq. (4). We provide estimation results for both options in our experiments.

**Problem setting.** Given a partial point cloud observation of a shape and the level-set parameters $\boldsymbol{\theta}$ representing the shape in its reference pose, we estimate the pose of the observation by optimizing the pose-dependent level-set parameters for the point cloud. This process follows the standard training procedures of the SDF network. During training, we maintain the reference level-set parameters $\boldsymbol{\theta}$ frozen. The pose parameters are trained using the distances of points to surface, *i.e.*, the $\mathcal{L}_{\text{dist}}^p$ loss in Eq. (7) for SDF reconstruction. We consider all points as samples on the surface of the shape.

**Pose initialization.** We define the rotation parameters using Euler angles $\boldsymbol{\omega} = (\alpha, \beta, \gamma)$. The pose estimation involves 3 parameters for rotation and 3 parameters for translation. In the context of SDF, we consider only small translations in the registration. Therefore, we always initialize the translation as $\mathbf{t} = \mathbf{0}$. For the rotation angles, we uniformly partition the space of $[0, 2\pi] \times [0, 2\pi] \times [0, 2\pi]$ into distinct subspaces, and initialize $\boldsymbol{\omega}$ with the centers of each subspace. This results in a total of $T^3$ initializations for $\boldsymbol{\omega}$, denoted as $\Omega = \left\{ \left( \frac{2\pi t_\alpha}{T}, \frac{2\pi t_\beta}{T}, \frac{2\pi t_\gamma}{T} \right) \big| t_\alpha, t_\beta, t_\gamma \in [T] \right\}$.

**Candidate Euler angles.** For each initialization with $\boldsymbol{\omega} \in \Omega$ and $\mathbf{t} = \mathbf{0}$, we compute the pose-dependent level-set parameters. Using the updated parameters, we predict the SDF values of each point in the partial point cloud, and compute the registration error $E_{\text{reg}}$ using $\mathcal{L}_{\text{dist}}^p$. This results in a

number of $T^3$ registration errors, denoted as $\{E_{\text{reg}}^i\}_{i=1}^{T^3}$. We sort these registration errors and select the candidate Euler angles $\Omega^* = \{\boldsymbol{\omega}_i\}_{i=1}^S$ corresponding to the top $S$ smallest registration errors.

**Pose estimation.** Given the candidate Euler angles $\Omega^*$, we employ Algorithm 2 to estimate the optimal pose. Specifically, for each candidate $\boldsymbol{\omega}_i$, we alternate between optimizing the initialized $\boldsymbol{\omega}$ and $\mathbf{t}$, each for $M$ iterations, over $N$ rounds. We record the optimized pose $(\hat{\boldsymbol{\omega}}_i, \hat{\mathbf{t}}_i)$ and registration loss $E_{\text{reg}}^i$ of each candidate $\boldsymbol{\omega}_i$. The optimal pose $(\hat{\boldsymbol{\omega}}_s, \hat{\mathbf{t}}_s)$ is determined as the pair resulting in the smallest registration loss $E_{\text{reg}}^s$. To enhance accuracy in practice, we continue optimizing $(\hat{\boldsymbol{\omega}}_s, \hat{\mathbf{t}}_s)$ by repeating steps 5-6 in Algorithm 2 until convergence. In our experiments, we set $T=15$, $S=20$, $N=20$, and $M=10$.

---

**Algorithm 2**

1: Compute candidate Euler angles $\Omega^* = \{\boldsymbol{\omega}_i\}_{i=1}^S$.
2: **for** $i = 1$ **to** $S$ **do**
3:     $\boldsymbol{\omega} \leftarrow \boldsymbol{\omega}_i, \mathbf{t} \leftarrow \mathbf{0}$.
4:     **for** $j = 1$ **to** $N$ **do**
5:         Freeze $\mathbf{t}$, optimize $\boldsymbol{\omega}$ for $M$ iterations.
6:         Freeze $\boldsymbol{\omega}$, optimize $\mathbf{t}$ for $M$ iterations.
7:     **end for**
8:     Record the optimized $(\hat{\boldsymbol{\omega}}_i, \hat{\mathbf{t}}_i)$ and $E_{\text{reg}}^i$.
9: **end for**
10: Get the index $s$ of the smallest loss in $\{E_{\text{reg}}^i\}_{i=1}^S$.
11: **return** $(\hat{\boldsymbol{\omega}}_s, \hat{\mathbf{t}}_s)$.

---

## 5 EXPERIMENT

ShapeNet (Chang et al., 2015) and Manifold40 (Hu et al., 2022) provide multi-category 3D shapes that are well-suited for geometric research in computer graphics and robotics. For the interested shape analysis, we construct Level-Set Parameter Data (LSPData) as continuous shape representations for these datasets. The two-stage training of HyperSE3-SDF in § 3.2 is adopted in this process. In the first stage of constructing the LSPData, we utilize 20 shapes from each class in ShapeNet and 7 shapes per class in Manifold40 to train the pose-dependent initialization $\boldsymbol{\mu}$. The point clouds of each shape for SDF are sampled from the shape surfaces in Manifold40 and sourced from previous work (Mescheder et al., 2019) in ShapeNet. We did not consider the loudspeaker category in ShapeNet due to the intricate internal structures of the shapes.

### 5.1 UNDERSTAND THE PARAMETER DATA

In the proposed dataset construction, we train HyperSE3-SDF for all shapes to create the LSPData with transformations for continuous shape analysis. To validate the approach, we conduct two extra experiments based on the plain SDF network to construct the LSPData, using different settings for $\boldsymbol{\mu}$. In the first experiment, we randomly initialize $\boldsymbol{\mu}$ and apply it to all shapes as Luigi et al. (2023). In the second, we learn $\boldsymbol{\mu}$ using the same training samples as those used for HyperSE3-SDF.

In this ablation study, we compare the constructed LSPData for five major categories of ShapeNet (Park et al., 2019a). We present the t-SNE embeddings (Van der Maaten & Hinton, 2008) of the data constructed by random and learned $\boldsymbol{\mu}$ in Fig. 3. We note that the embeddings for our HyperSE3-SDF in pose $(\mathbf{I}, \mathbf{0})$ are similar to Fig. 3(b). It can be seen that the learned $\boldsymbol{\mu}$ significantly enhances shape semantics compared to the randomly initialized counterpart. Table 1 compares the shape classification results of the different LSPData under different rotation setups with the same baseline encoder. '$(\mathbf{I}, \mathbf{0})$' represents testing with data in reference poses. '$z$/SO(3)' indicates training with data rotated around $z$ axis but testing on data arbitrarily rotated in SO(3) (Esteves et al., 2018). We obtain the transformed LSPData of plain SDF by applying Eq. (4). Notably, the transformed LSPData from HyperSE3-SDF surpasses those from Euclidean computations, while

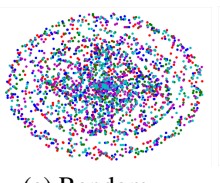
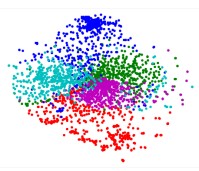

(a) Random $\boldsymbol{\mu}$.      (b) Learned $\boldsymbol{\mu}$.

Table 1: Shape semantics of different LSPData.

| Network | plain SDF | | HyperSE3-SDF |
|---|---|---|---|
| Initialization ($\boldsymbol{\mu}$) | random | learned | learned |
| $\Delta\boldsymbol{\theta}$ $(\mathbf{I}, \mathbf{0})$ | 41.77 | **97.0** | 96.79 |
| $\Delta\boldsymbol{\theta}$ $(z/\text{SO}(3))$ | - | 86.73 | **95.76** |
| $\boldsymbol{\theta}$ $(z/\text{SO}(3))$ | - | 75.48 | **80.39** |

Figure 3: t-SNE embeddings of level-set parameter data obtained with different $\boldsymbol{\mu}$, randomly initialized in (a) and learned in (b). Table 1 compares the shape classification accuracy of differently constructed LSPData, under different rotation setups. $(\mathbf{I}, \mathbf{0})$ represents data in reference poses, $z$ indicates data rotated around the vertical axis, and SO(3) stands for data rotated randomly.

the normalized LSPData $\Delta\boldsymbol{\theta}$ outperforms the unnormalized $\boldsymbol{\theta}$. We also conduct ablation studies to identify the optimal pose-dependent subset within the level-set parameters for the hypernetwork to generate. Details can be found in the appendix.

## 5.2 SHAPE CLASSIFICATION AND RETRIVAL

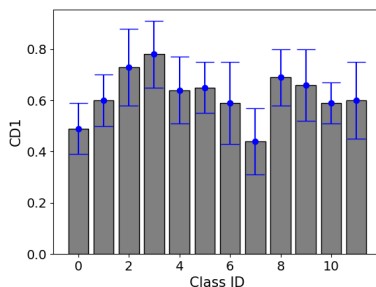

We apply the level-set parameters to semantic analysis on ShapeNet and Manifold40. For time concern, a maximum of 2000 shapes are reconstructed for each class in ShapeNet. We filter the continuous shapes based on their Chamfer distances to ensure surface quality. This results in a dataset of 17,656 shapes for ShapeNet and 10,859 for Manifold40. Figure 4 plots the mean and standard deviation of Chamfer distance (CD1) for each class in ShapeNet, illustrating the surface quality represented by our LSPData. Additionally, the mean and standard deviation of CD1 for all classes in Manifold40 are 0.7102 and 0.1443, respectively. Note that our chamfer distances are

Figure 4: Surface quality of ShapeNet.

averaged across multiple random poses due to the introduced surface transformation. Besides, our training set in ShapeNet comprises 200 shapes per class, while in Manifold40, it contains 50% of the shapes per class, up to 200 shapes.

Table 2: Shape classification with discrete and continuous representations on ShapeNet.

| Method | pose | OA | plane | bench | cab | car | chair | disp | lamp | rifle | sofa | table | phone | vessel |
|---|---|---|---|---|---|---|---|---|---|---|---|---|---|---|
| PointNet (Qi et al., 2017a) | | 27.34 | 17.01 | 33.16 | 27.41 | 13.87 | 29.83 | 21.88 | 35.28 | 61.11 | 12.36 | 11.74 | 22.61 | 38.13 |
| DGCNN (Wang et al., 2019d) | z/SO(3) | 24.41 | 81.76 | 4.82 | 19.31 | 43.75 | 13.60 | 4.28 | 59.87 | 0.00 | 4.09 | 4.65 | 8.38 | 22.16 |
| VN-DGCNN (Deng et al., 2021) | | 90.40 | 97.01 | 81.84 | 93.35 | 98.35 | 86.98 | 83.62 | 89.05 | 96.17 | 89.39 | 80.95 | 92.95 | 93.13 |
| PaRI-Conv (Chen & Cong, 2022) | | 91.94 | 97.65 | **83.69** | **93.46** | **99.45** | 90.66 | 86.92 | 93.48 | 98.51 | **92.23** | 79.32 | 94.68 | 91.94 |
| **LSPData ($\Delta\theta$)** | | **93.03** | **98.36** | 83.09 | 89.10 | 98.70 | **94.65** | **92.79** | **93.65** | **98.51** | 89.31 | **86.63** | **94.95** | 94.78 |
| PointNet (Qi et al., 2017a) | | 92.48 | 97.60 | 86.46 | 95.02 | 98.56 | 94.07 | 90.59 | 87.37 | 98.15 | 91.81 | 81.68 | 91.76 | 94.18 |
| PointNet++ (Qi et al., 2017c) | (I, 0) | 93.36 | 98.36 | 84.87 | 94.70 | 99.04 | 92.07 | 90.95 | 92.98 | 99.15 | 90.64 | 86.26 | 96.01 | 94.85 |
| **LSPData ($\Delta\theta$)** | | 93.22 | 98.59 | 84.54 | 91.07 | 99.52 | 94.39 | 93.03 | 92.73 | 99.43 | 91.14 | 83.90 | 95.48 | 93.51 |

**ShapeNet.** We conduct pose-related semantic analysis on ShapeNet using rotation setups of $z$/SO(3) and $(\mathbf{I}, \mathbf{0})$. Table 2 demonstrates the feasibility of shape analysis based on level-set parameters *without dependence on* point clouds and meshes (Luigi et al., 2023). We compare our results to point cloud-based methods. It can be seen that the proposed continuous representation shows comparable performance to point clouds in shape classification of pose $(\mathbf{I}, \mathbf{0})$. In the challenging setup of $z$/SO(3), our encoder-based network for continuous shapes outperforms the rotation-equivariant networks for point clouds. This is attributed to the separation of pose-dependent and pose-independent parameters in the continuous representations. In addition, we observe that our classification network converges rapidly within a few epochs, consistent with the findings in (Dupont et al., 2022).

**Manifold40.** We also compare the effectiveness of LSPData in shape classification and retrieval with point clouds under the rotation setup SO(3)/SO(3), abbreviated as SO(3) in Table 3. For retrieval, we extract features before the classifier to match shapes in the feature space via Euclidean distances. We eval-

Table 3: Shape classification and retrieval on Manifold40.

| Method | Pose | classification | | retrieval (mAP) | | |
|---|---|---|---|---|---|---|
| | | OA | mAcc | top1 | top5 | top10 |
| PointNet | | 79.33 | 70.63 | 63.99 | 59.88 | 56.84 |
| DGCNN | | 82.70 | 74.98 | 70.82 | 67.14 | 64.85 |
| VN-DGCNN | SO(3) | 84.61 | 78.25 | 80.02 | 77.13 | 74.99 |
| PaRI-Conv | | 85.14 | 76.44 | 82.51 | 80.39 | 78.82 |
| **LSPData ($\Delta\theta$)** | | **86.89** | **79.36** | **85.19** | **83.78** | **82.75** |

uate the retrieval performance using mean Average Precision (mAP) alongside the top-1/5/10 recalls. It can be noticed that our results based on continuous shapes using LSPData outperform those for point clouds. We note that the performance gap between Manifold40 and ShapeNet is mainly attributed to the restricted number of shapes in certain classes of Manifold40.

## 5.3 OBJECT POSE ESTIMATION

**Data preparation.** To prepare the partial point clouds for pose estimation, we select three categories from ShapeNet: *airplane*, *car*, and *chair*, with 10 shapes utilized in each category. For every shape, we create 10 ground-truth transformations with random rotations in the range of $[0, 2\pi]$ and translations in the range of $[-0.1, 0.1]$. For each transformed shape, we create the partial point cloud from its full point cloud representation, using hidden point removal (Katz et al., 2007). This results in 300 pairs that covers partial and full point clouds with limited overlaps and large rotations. We

Table 4: Optimization-based registration for pose estimation.

| Method | $\sigma=0$ | | $\sigma=0.01$ | | $\sigma=0.03$ | | $\sigma=0.01$, 30% outlier | |
|---|---|---|---|---|---|---|---|---|
| | RRE↓ | RTE↓ | RRE↓ | RTE↓ | RRE↓ | RTE↓ | RRE↓ | RTE↓ |
| ICP (Besl & McKay, 1992) | 134.08 | 27.71 | - | - | - | - | - | - |
| FGR (Zhou et al., 2016) | 105.88 | 19.91 | - | - | - | - | - | - |
| TEASER++ (Yang et al., 2020) | 12.98 | 4.91 | 126.82 | 60.26 | - | - | - | - |
| **Proposed** (V1) | 0.12 | 0.16 | **0.21** | 0.32 | **1.36** | 1.65 | 1.35 | 0.55 |
| **Proposed** (V2) | **0.06** | **0.12** | 0.78 | **0.29** | 1.38 | **1.63** | **1.25** | **0.54** |

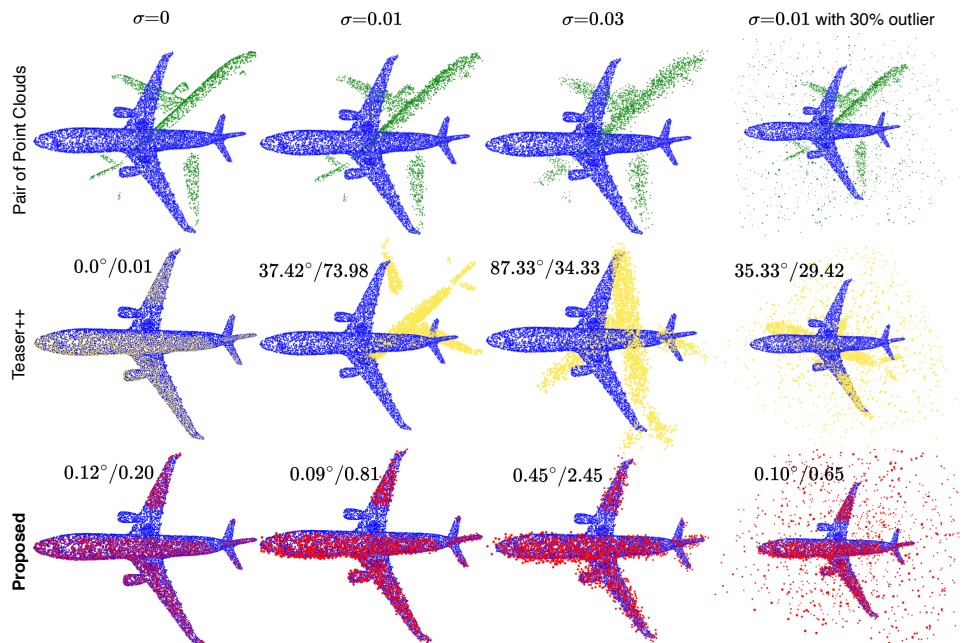

Figure 5: Registration of Teaser++ and ours on an *airplane*. RRE/RTE metrics are shown, with RTE scaled by $\times 100$. The ground-truth Euler angles in this case are $\boldsymbol{\omega}=(202.97°, 352.18°, 256.89°)$.

introduce different levels of noise, $\sigma \in \{0.01, 0.03\}$, and an outlier ratio of 30% to the partial point clouds for further challenges.

Given challenges posed by the absence of training data, we focus our comparison on optimization-based registration methods rather than deep learning-based approaches. For readers interested in the deep learning alternatives, we provide the performance of the pretrained GeoTransformer (Qin et al., 2023) on our data in the appendix. Among the optimization-based methods, we compare the estimation quality of our method, Proposed V1 and V2 (see § 4.2), with ICP (Besl & McKay, 1992), FGR (Zhou et al., 2016), and TEASER++ (Yang et al., 2020) in Table 4. Their performance is evaluated based on the relative rotation/translation errors (RRE/RTE). See the appendix for definitions of the two metrics. RRE is reported in degrees, and RTE is scaled by $\times 100$. Notably, the proposed method effectively estimates poses with arbitrary rotations from partial-view point clouds, even in the presence of significant noise and outliers. Visualized examples are provided in Fig. 5 and Fig. 6. We notice that while TEASER++ recovers most poses in the clean data, its performance drops with noise and outliers. ICP and FGR struggle with estimating large rotations even for clean point clouds. If a method fails in simpler settings, we cease testing it on more challenging data. The proposed method takes $\sim 50$ seconds to estimate the poses accurately.

# 6 CONCLUSION

This paper extends shape analysis beyond traditional 3D data by introducing level-set parameters as a continuous and numerical representation of 3D shapes. We establish shape correlations in the non-Euclidean parameter space with learned SDF initialization. A novel hypernetwork is proposed to transform the shape surface by modifying a subset of level-set parameters according to rotations and translations in SE(3). The resulting continuous representations facilitate semantic shape analysis in SO(3) compared to the Euclidean-based transformations of continuous shapes. We also demonstrate the efficacy of level-set parameters in geometric shape analysis with pose estimation. The SDF

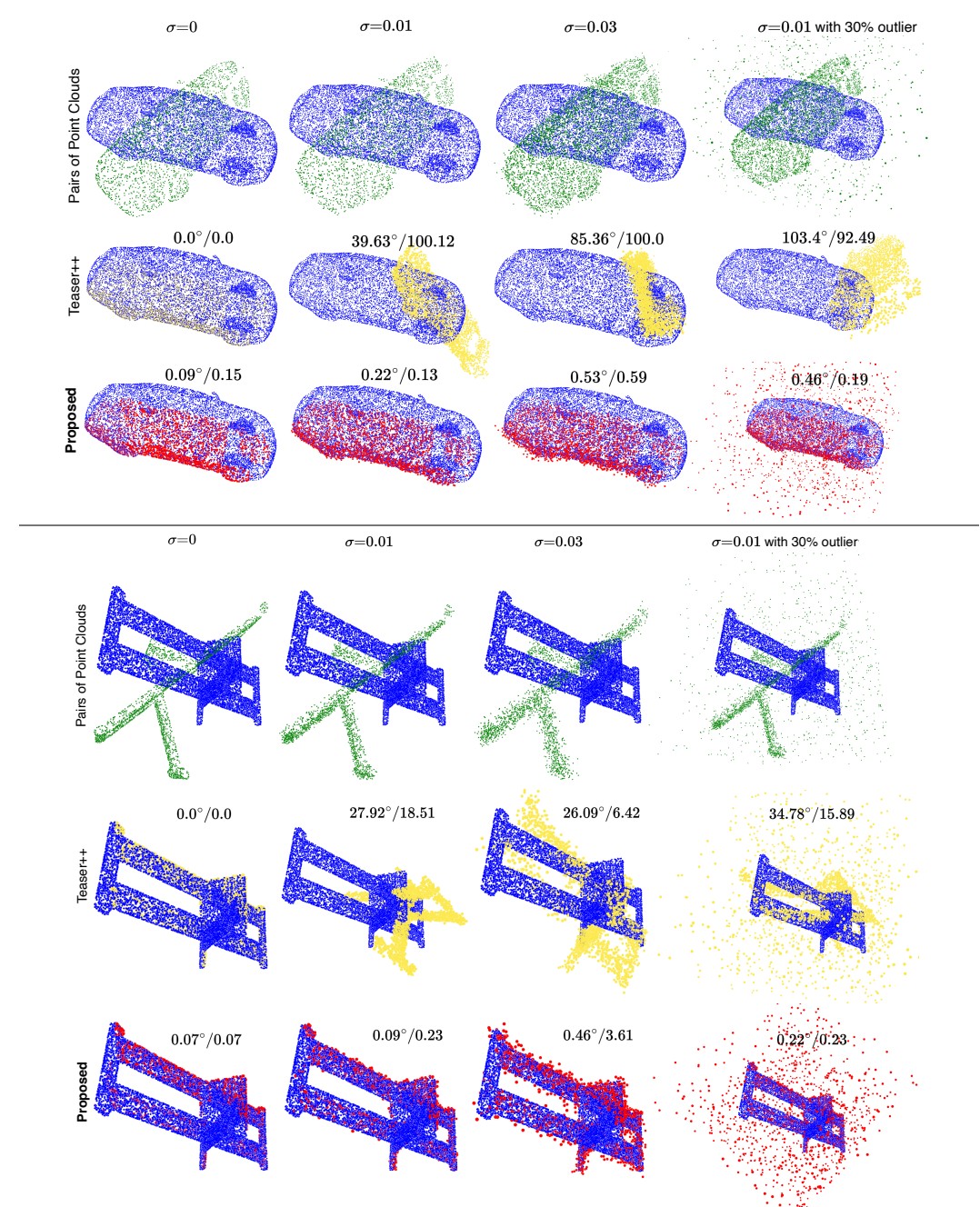

Figure 6: Registrations of Teaser++ and our method on a *car* and a *chair*.

reconstruction loss is leveraged to optimize the pose parameters, yielding accurate estimations for poses with arbitrarily large rotations, even when the data contain significant noise and outliers.

**Limitations.** Level-set parameters summarize the overall topology of 3D shapes and can be applied to global feature learning of continuous shapes. However, they are not suitable for learning local features of 3D shapes, as there is no correspondence between the local structures of the shape and subsets of the level-set parameters. Researchers have explored local modulation vectors (Bauer et al., 2023) to improve the image classification performance of continuous representations, but with limited success. Mixtures of neural implicit functions (You et al., 2024) may offer enhanced local encoding in the continuous representations. Besides, distinguishing the level-set parameters into pose-dependent and independent subsets, and making them learnable from arbitrarily posed point clouds, could be important for continuous shape representations and analysis in the future. Extending continuous representations from shapes to real-world surfaces also merits exploration.

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

# A THE SDF NETWORK

## A.1 TENSORS OF LEVEL-SET PARAMETERS

We show the 8-layer SDF network with skipping concatenation in Fig. A. The resulting level-set parameters $\Theta_1, \Theta_2, \Theta_3$ have dimensions $256 \times 4$, $6 \times 256 \times 257$, and $1 \times 257$, respectively. We use the proposed hypernetwork $h_\phi$ to generate the first layer parameters for surface transformation.

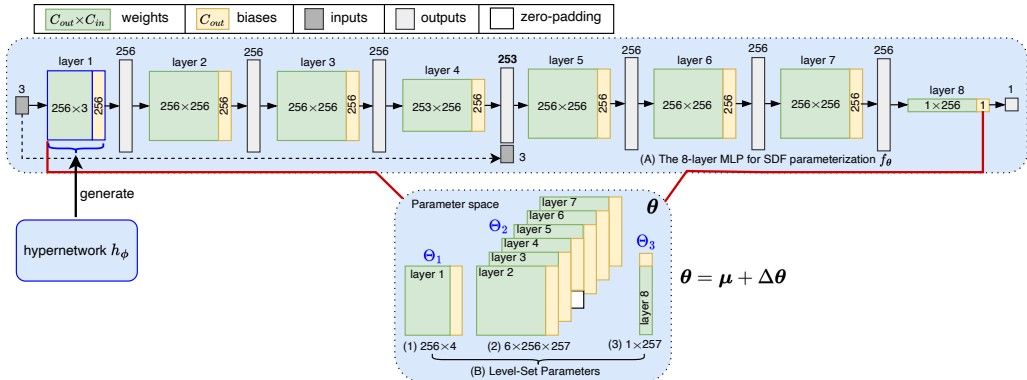

Figure A: The SDF network and its resulting level-set parameters for shape representation.

## A.2 UNSUPERVISED SDF RECONSTRUCTION LOSS

Let $\mathcal{X}_p$ and $\mathcal{X}_n$ be point clouds sampled on and off the surface. $\hat{\mathbf{n}}(\mathbf{x})$ and $\mathbf{n}(\mathbf{x})$ be the estimated normal and ground truth normal, respectively. We use the loss function $\mathcal{L}_{\text{SDF}}$ in Eq. (2) consisting of four objectives $\mathcal{L}_{\text{dist}}^p, \mathcal{L}_{\text{dist}}^n, \mathcal{L}_{\text{eik}}, \mathcal{L}_{\text{norm}}^p$ for SDF reconstruction. They each are computed as

$$\mathcal{L}_{\text{dist}}^p = \sum_{\mathbf{x} \in \mathcal{X}_p} \|f_{\boldsymbol{\theta}}(\mathbf{x})\|_1, \tag{7}$$

$$\mathcal{L}_{\text{dist}}^n = \sum_{\mathbf{x} \in \mathcal{X}_n} \exp(-\rho \|f_{\boldsymbol{\theta}}(\mathbf{x})\|_1), \rho \gg 1, \tag{8}$$

$$\mathcal{L}_{\text{eik}} = \sum_{\mathbf{x} \in \mathcal{X}_p \bigcup \mathcal{X}_n} (\|\nabla f_{\boldsymbol{\theta}}(\mathbf{x})\|_2 - 1)^2, \tag{9}$$

$$\mathcal{L}_{\text{norm}}^p = \sum_{\mathbf{x} \in \mathcal{X}_p} \|1 - \langle \hat{\mathbf{n}}(\mathbf{x}), \mathbf{n}(\mathbf{x}) \rangle\|_1 + \|\hat{\mathbf{n}}(\mathbf{x}) - \mathbf{n}(\mathbf{x})\|_1. \tag{10}$$

We use $\mathcal{L}_{\text{SDF}}$ in the first stage of dataset construction to learn the shard parameters $\boldsymbol{\mu}$. In the second stage, we add two regularization terms to the $\mathcal{L}_{\text{SDF}}$ and train the parameters $\{\Delta Y_{ij}^{mn}\}$ and $\{\Delta a_\ell^{mn}\}$ associated with shape instances with the loss below,

$$\mathcal{L} = \mathcal{L}_{\text{SDF}} + \frac{\lambda_{\text{reg}}}{K} \Big( \sum_{m,n,i,j} |\Delta Y_{ij}^{mn}| + \sum_{\ell,m,n} |\Delta a_\ell^{mn}| \Big). \tag{11}$$

The two extra terms encourage $\Delta \boldsymbol{\theta}$ to be close to zero. $\lambda_{\text{reg}}$ is a hyperparameter, and $K$ denotes the total number of training parameters in $\{\Delta Y_{ij}^{mn}\}$ and $\{\Delta a_\ell^{mn}\}$. The L1 Loss is denoted by $|\cdot|$.

# B SDF-BASED SURFACE TRANSFORMATION

## B.1 GEOMETRIC SDF INITIALIZATION WITH THE HYPERNETWORK

Figure B illustrates the proposed hypernetwork. We sample from the standard normal distribution to initialize all entries in each latent matrix $\mathbf{Y}^{mn}$. This matrix is combined with the pose-dependent coefficient matrix $\mathbf{B}^{mn}$ to compute a vector $\mathbf{z}^{mn}$ that follows normal distributions. Specifically, we

Figure B: Hypernetwork for surface transformation. The hypernetwork $h_\phi$ utilizes a 4-layer MLP with output channels of 256, 256, 256, and 16384 to compute a compact set of pose-dependent coefficient matrices $\{\mathbf{B}^{mn}\}$. They are combined with the latent matrices $\{\mathbf{Y}^{mn}\}$ to compute the vectors $\{\mathbf{z}^{mn}\}$, which are then normalized into $\{\hat{\mathbf{z}}^{mn}\}$ to satisfy the standard normal distribution. In the final step, two branches of fully connected layers, FC1 and FC2, take each $\mathbf{z}^{mn}$ and $\hat{\mathbf{z}}^{mn}$ as input to generate a pose-dependent bias and weight for the first SDF layer, respectively.

formulate each element $z_j$ in $\mathbf{z}$ as

$$z_j = \sum_{i=1}^{I} B_{ij} Y_{ij}, \; j \in [J], \tag{12}$$

which is a linear combination of variables in the $j$th column of $\mathbf{Y}$. The superscripts $m, n$ are omitted.

Let $\mathcal{N}(0,1)$ be the standard normal distribution. For any random variable $Z = \sum_i \beta_i Y_i$ with $Y_i \sim \mathcal{N}(0,1) \; \forall i$, its expectation and variance are

$$\mathbb{E}(Z) = \sum_i \beta_i \mathbb{E}(Y_i) = 0. \tag{13}$$

$$\mathrm{Var}(Z) = \sum_i \beta_i^2 \mathrm{Var}(Y_i) = \sum_i \beta_i^2. \tag{14}$$

The normalized variable $\hat{Z} = \frac{Z}{\sqrt{\sum_i \beta_i^2}} \sim \mathcal{N}(0,1)$ [78]. We normalize each element in $\mathbf{z}$ as $\hat{z}_j = \frac{z_j}{\sqrt{\sum_i B_{ij}^2}}$ to obtain a normalized vector $\hat{\mathbf{z}}$ where $\hat{z}_j \sim \mathcal{N}(0,1)$.

Given the normalized vector $\hat{\mathbf{z}}$ which strictly follows a standard normal distribution, we design two different branches, FC1 and FC2, in the last layer of $h_\phi$ to initialize the weight and bias parameters in $\{\theta_1^{mn}\}$ according to SAL [58]. We note that the approximate Gaussian Process in [79] does not help in generating outputs that follow the standard normal distribution for the required initializations.

**Initialization to zero.** We use FC1 to generate the SDF biases in $\{\theta_1^{mn}\}$. It takes each $\mathbf{z}^{mn}$ as inputs. We initialize all weights and bias of FC1 as zeros to ensure that the generated SDF biases start at zero.

**Initialization to normal distributions.** The SDF weights should be initialized following certain normal distributions $\mathcal{N}(\mu, \sigma^2)$, where $\mu = 0$ and $\sigma = \sqrt{2/256}$ in our SDF network. We introduce FC2, which takes $\hat{\mathbf{z}}^{mn}$ as inputs to generate $\theta_1^{mn}$. Let $\mathbf{w} \in \mathbb{R}^J$ be the neural weights of FC2. The detailed computation of $\theta_1^{mn}$ from $\hat{\mathbf{z}}^{mn}$ is given by

$$\theta_1^{mn} = \mu + \sigma \frac{\langle \mathbf{w}, \hat{\mathbf{z}}^{mn} \rangle}{\langle \mathbf{w}, \mathbf{w} \rangle}. \tag{15}$$

$\langle \cdot, \cdot \rangle$ represents the scalar products between two vectors. Note that FC2 does not require biases.

The proposed hypernetwork emphasizes importance of the first SDF layer while allowing the SDF parameters of the other layers to be shared across different poses, substantially reducing the neural parameter size of $h_\phi$. The introduction of the latent matrices $\{\mathbf{Y}^{mn}\}$ is important as it enables the hypernetwork to satisfy the geometric initializations of SDF network. Note that parameters of the unconditioned layers (layer 2-8) in SDF are initialized the same as in SAL [58].

## B.2 Euclidean-based SDF Transformation

3D shapes can be transformed in the parameter space by applying Euclidean transformation to their level-set parameters in the reference poses. Let $\mathbf{W}$ and $\mathbf{b}$ be the neural weights and biases that

Table A: The surface quality of different methods in implicit surface transformations.

| Method | Baseline | | Baseline++ | | HyperSE3-SDF | | **HyperSE3-SDF Dataset** | |
|---|---|---|---|---|---|---|---|---|
| | CD1 ↓ | NC ↑ | CD1 ↓ | NC ↑ | CD1 ↓ | NC ↑ | CD1 ↓ | NC ↑ |
| #epochs/runtime | 10000/1 hour | | | | | | 500/4 minutes | |
| airplane | 2.83±3.71 | 0.94±0.05 | 0.54±0.06 | 0.98±0.00 | **0.48±0.02** | **0.99±0.00** | 0.53±0.06 | 0.99±0.00 |
| | 1.44±0.77 | 0.93±0.03 | 0.90±0.31 | 0.96±0.01 | **0.48±0.07** | **0.98±0.00** | 0.48±0.05 | 0.98±0.00 |
| car | 1.50±0.50 | 0.95±0.01 | 1.20±0.41 | 0.96±0.01 | **0.73±0.18** | **0.98±0.00** | 0.66±0.18 | 0.98±0.00 |
| | 0.97±0.12 | 0.95±0.00 | 1.33±0.42 | 0.93±0.01 | **0.72±0.03** | **0.96±0.00** | 0.74±0.04 | 0.96±0.00 |
| chair | 11.51±2.72 | 0.71±0.09 | 8.77±5.49 | 0.78±0.14 | **0.61±0.07** | **0.98±0.00** | 0.57±0.06 | 0.99±0.00 |
| | 10.21±3.37 | 0.75±0.14 | 2.57±3.35 | 0.92±0.11 | **0.55±0.07** | **0.98±0.00** | 0.54±0.05 | 0.98±0.00 |
| lamp | 1.33±0.56 | 0.95±0.02 | 1.05±0.32 | 0.97±0.01 | **0.62±0.08** | **0.98±0.00** | 0.56±0.05 | 0.98±0.00 |
| | 1.18±0.33 | 0.97±0.01 | 0.96±0.23 | 0.97±0.01 | **0.70±0.24** | **0.98±0.01** | 0.64±0.14 | 0.98±0.00 |
| table | 1.01±0.27 | 0.96±0.01 | 0.83±0.22 | 0.97±0.01 | **0.62±0.10** | **0.98±0.00** | 0.55±0.03 | 0.98±0.00 |
| | 0.80±0.17 | 0.96±0.01 | 0.98±0.20 | 0.95±0.01 | **0.62±0.10** | **0.97±0.00** | 0.60±0.09 | 0.98±0.00 |

apply to the input coordinates $\mathbf{x}$, *i.e.*, $\mathbf{Wx} + \mathbf{b}$. For the surface transformed with $\mathbf{R}, \mathbf{t}$, resulting in $\mathbf{y} = \mathbf{Rx} + \mathbf{t}$ in the Euclidean space, the related parameters can be modified as

$$\mathbf{W}' = \mathbf{WR}^{-1}, \mathbf{b}' = -\mathbf{WR}^{-1}\mathbf{t} + \mathbf{b}. \qquad (16)$$

It is obtained by replacing the $\mathbf{x}$ in $\mathbf{Wx} + \mathbf{b}$ with $\mathbf{x} = \mathbf{R}^{-1}(\mathbf{y} - \mathbf{t})$. However, this Euclidean-based SDF transformation yields level-set parameter data that require more data augmentations for shape classification and retrieval in SO(3), similar to point cloud data.

## C    IMPLEMENTATION DETAILS

**Hypernetwork $h_\phi$.** In the first stage of our dataset construction, we train HyperSE3-SDF with a batch size of 50 for 50000 epochs. We utilize the Adam Optimizer [80] with an initial learning rate of 0.001, which exponentially decays at a rate of $\gamma$=0.998 every 30 epochs to train the model. This training process takes 3 days on a GeForce RTX 4090 GPU.

**Sampling Transformations.** We consider Euler angles in the range of $[0, 2\pi]$ and translations in the subspace of $[-0.1, 0.1]^3$ in the context of SDF. For dataset construction, we randomly sample 150 pairs of $(\mathbf{R}, \mathbf{t})$, including the reference pose $(\mathbf{I}, \mathbf{0})$, which are fixed and shared across all epochs. Additionally, in each epoch, we randomly sample another 150 $(\mathbf{R}, \mathbf{t})$ on-the-fly. These fixed and on-the-fly transformations are collectively utilized to train the HyperSE3-SDF network. This strategy facilitates satisfactory convergence and generalization of the model.

**Augmentation of Level-Set Parameters.** In the semantic shape analysis with level-set parameters, we apply scaling operations to the level-set parameters using normal distributions with a major sigma of $\sigma_1$=0.2 and a minor sigma of $\sigma_2$=0.05 for small perturbations of different $\theta$. We also perturb the level-set value $c$=0 using normal distributions with $\sigma_c$=0.1, and apply positional encoding to $c$, denoted as PE($c$). The resulting PE($c$) is concatenated with $\Theta_3$ for feature extraction from the corresponding branch. We set the level of PE($c$) to 10 for ShapeNet and 16 for Manifold40. In the transformer layer, we apply dropout with a rate of 0.3.

**Other Details.** We use 500 shapes from each of the five categories: *airplane*, *chair*, *lamp*, *sofa*, *table*, in the experiments of § 5.1.

## D    POSE-DEPENDENT PARAMETERS IN $\boldsymbol{\theta}$

To verify the choice of our HyperSE3-SDF on pose-dependent level-set parameters, we introduce two variants: Baseline and Baseline++, which condition different subsets of level-set parameters on rotations and translations. Specifically, Baseline utilizes the hypernetwork to generate biases of the first SDF layer, while Baseline++ generates biases of all layers with the hypernetwork. We compare their performance to the proposed hypernetwork, which generates weights and biases of the first SDF layer. In this study, we train the different networks without learned initializations.

We evaluate surface reconstruction using L1 Chamfer Distance (CD1×100) and Normal Consistency (NC). To assess overall surface quality across various transformations, we randomly sampled 50 poses, calculating the mean and deviation of these metrics to report performance. Table A summarizes the results, showing that the proposed hypernetwork significantly outperforms Baseline and Baseline++. For reference, we provide the corresponding shape quality in our two-stage constructed dataset. The training time for a shape in the second stage of dataset construction is 4 minutes, which is significantly less than the 1 hour required for a hypernetwork without learned initializations.

In the Fig. C below, we visualize the transformed surfaces of an airplane and a chair using different hypernetworks. The first column represents the reference pose, while the subsequent columns display randomly transformed surfaces. This visualization reaffirms the superiority of HyperSE3-SDF.

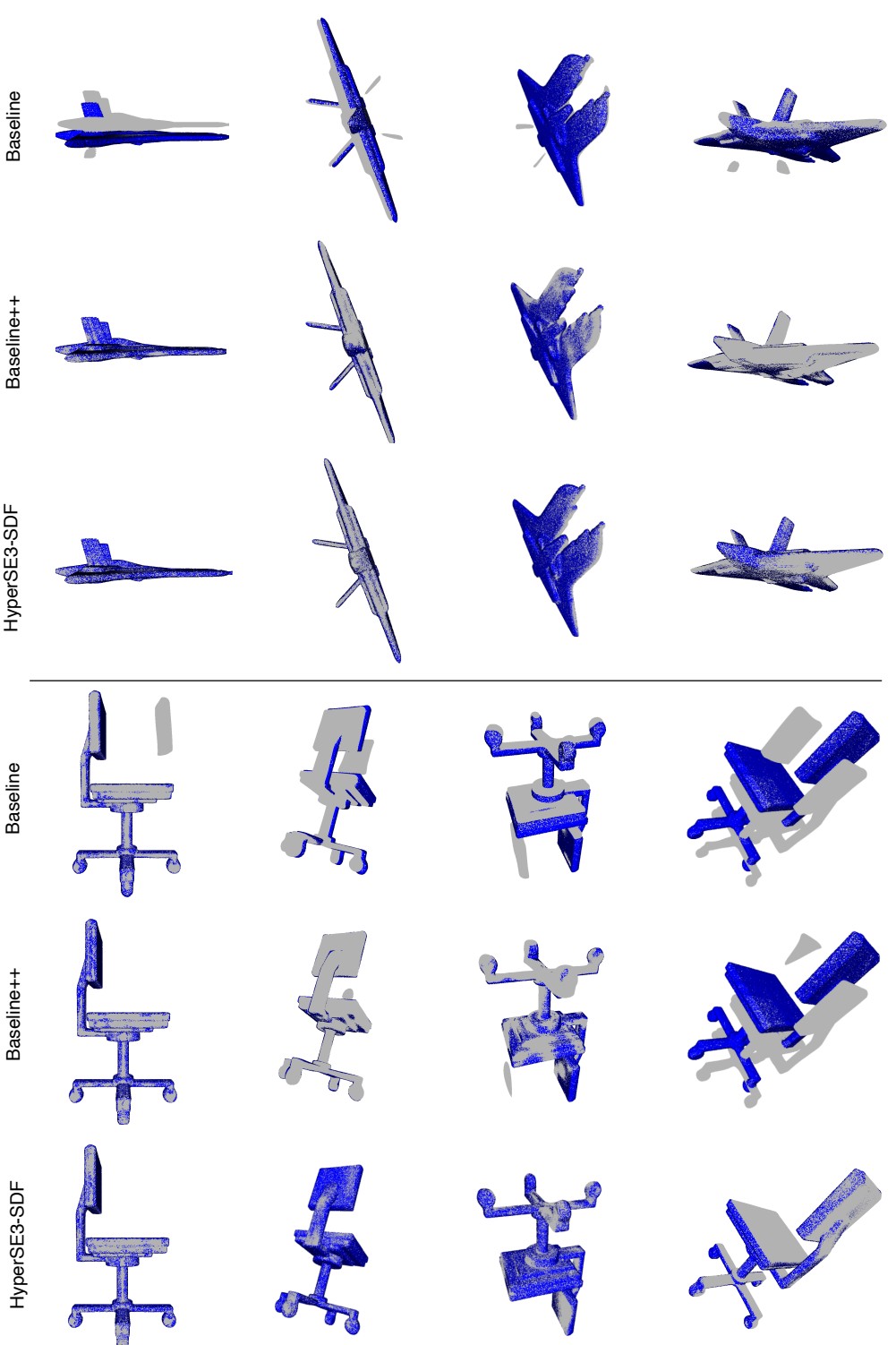

Figure C: Comparison of different hypernetworks at transforming the continuous shape surfaces. We show the transformed point cloud with blue dots on top of the shape surface. The proposed HyperSE3-SDF performs the best, while Baseline++ outperforms Baseline.

# E  POSE ESTIMATION

**Evaluation Metrics.** Let $\mathbf{R}, \mathbf{t}$ be the ground-truth rotation and translation, respectively, while $\hat{\mathbf{R}}, \hat{\mathbf{t}}$ be their estimated counterparts. We evaluate the pose estimation quality using Relative Rotation Error (RRE) and Relative Translation Error (RTE), calculated as follows:

$$\text{RRE} = \arccos\left(\frac{\text{tr}(\hat{\mathbf{R}}^\mathsf{T}\mathbf{R}) - 1}{2}\right), \ \text{RTE} = \|\hat{\mathbf{t}} - \mathbf{t}\|_2. \tag{17}$$

We show in Fig. D distributions of the ground-truth Euler angles in our 300 point cloud pairs. It can be seen that the angles vary from $0°$ to $360°$. Table B reports the results of GeoTransformer (Qin et al., 2023) on ModelNet and our data, using their model pretrained on ModelNet40. It can be seen that the method performs well for small rotations with Euler angles in the range of $[0°, 45°]$ on ModelNet but significantly degrades with larger rotations. On our data, its performance is unsatisfactory even for small rotations due to the domain-gap.

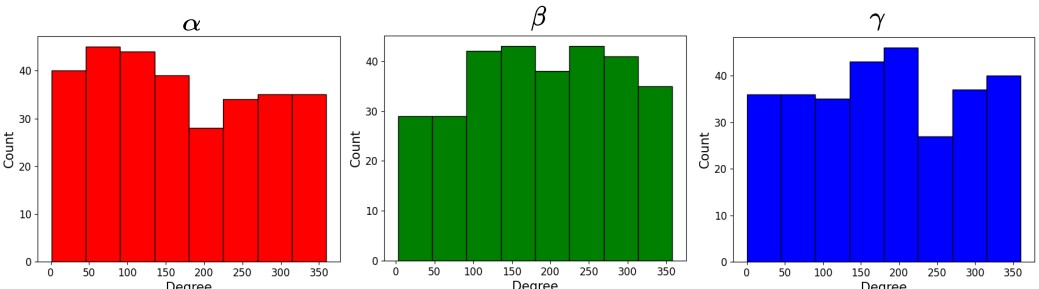

Figure D: Distributions of Euler Angles for the ground-truth rotations.

Table B: GeoTransformer on ModelNet and our data.

| Data | $\boldsymbol{\omega} \in [0°, 45°]^3$ | | $\boldsymbol{\omega} \in [0°, 90°]^3$ | | $\boldsymbol{\omega} \in [0°, 180°]^3$ | | $\boldsymbol{\omega} \in [0°, 360°]^3$ | |
|---|---|---|---|---|---|---|---|---|
| | RRE↓ | RTE↓ | RRE↓ | RTE↓ | RRE↓ | RTE↓ | RRE↓ | RTE↓ |
| ModelNet | 1.01 | 0.7 | 47.65 | 19.10 | 130.33 | 37.10 | 125.92 | 36.20 |
| Ours | 56.82 | 38.40 | 85.89 | 56.60 | 132.04 | 58.60 | 130.54 | 58.0 |

