# OpenReview forum: "Level-Set Parameters: Novel Data for 3D Shape Analysis"
_ICLR.cc/2025/Conference — ICLR 2025 Conference Withdrawn Submission_

### Official Review · Reviewer_8TmQ · 2024-10-16

**Soundness:** 3
**Presentation:** 3
**Contribution:** 2
**Rating:** 5
**Confidence:** 3

**Summary:**

This paper train a deepsdf where shape codes are explicitly split into shape parameters and pose parameters. In order to make this split, it share shape parameters mu for same objects with different poses. To implement the pose-dependent generation, it introduces a hypernetwork that generates the first-layer MLP in DeepSDF given R,t.

**Strengths:**

The technique is sound and the experiments proves its solidness. I also appreciate the authors for sharing many details for the network design and parameters. The paper is well written.

**Weaknesses:**

I am a little concerned about the novelty and the meaning of doing so.

First, pose-dependent shape generation seems to be well-explored, not only for objects but also for human faces/bodies. I believe a survey of these works is necessary. Also, what are the novelties for this work given existence of pose-dependent generative networks?

Second, the authors may argue that the pose network can be clearly handled separately. However, with more and more shapes, can the network really handle them separately? For example, given fixed shape parameters and random pose input, it is hard to justify that the network will produce shapes with variance only in poses rather than shapes. Even if it is explicitly enforced during training, can the network inherit this capability during inference (especially when data scale become large).

Third, as a decoder-only network, it seems that there is no possibility for it to handle novel shapes, which should be considered a big limitation. Therefore, it is hard for me to appreciate its application value.

**Questions:**

What is the difference of technique / novelty of this paper compared to pose-guided generative networks for human bodies/faces?

Are there any other similar works for doing pose-guided rigid shape generation?

---

### Official Review · Reviewer_T53i · 2024-10-20

**Soundness:** 2
**Presentation:** 4
**Contribution:** 2
**Rating:** 5
**Confidence:** 4

**Summary:**

The work proposes a novel idea using the level-set parameters instead of traditional data modality for shape analysis, including shape classification, retrieval and pose estimation. This idea is implemented by formulating the parameters as pseudo-normal distribution to maintain the correlations across data.

**Strengths:**

1 I like the constraints the work introduces in the Hypernetwork. Without the constraints, the correlations among the shapes may disappear in high-dimension space and are not suitable for shape analysis.

2 The work demonstrates how to encode the proposed new data modality for tasks. This new data modality of the dimension is huge since the level-set parameters are network weights outputted from Hypernetwork. Hence, the encoder design for the data modality is significant.

**Weaknesses:**

1 Need to construct level-set parameters for each shape before shape analysis.

2 Although the performance is improved, I feel that the proposed data modality complicates the task. Take the shape classification task as an example: traditional methods can directly take conventional data as input. However, with the proposed method, a dataset construction step is required. This involves using a Hypernetwork to generate the weights of another neural network (SDF), which is trained on traditional data modalities such as points and normals. Then, a specialized neural network (semantic learning network) needs to be designed to handle the large dimensionality of parameters as input for classification training. This reduces the flexibility in network design choices. Also, if users want to add new data or a new shape, they may need to retrain the Hypernetwork and SDF network. If so, this is time-consuming.

3 The supervised methods in Table 2 are just PointNet and PointNet++. Although these methods are famous, they are also old methods. Please consider including more recent works.

4 Table 2 should also consider reporting the number of parameters of each model for a fair comparison.

5 PaRI-Conv's performance is as competitive as the proposed method in Tables 2 and 3.  PaRI-Conv even outperforms the proposed method in some categories in Table 2.

6 I do not like this statement "In addition, we observe that our classification network converges rapidly within a few epochs, consistent with the findings in (Dupontetal.,2022)." The time of dataset construction should be taken into account.

7 I feel it is too ambitious to propose a new data modality. If the focus of the paper is on a new data modality, I would expect more versatile shape analysis beyond just classification, pose estimation, and retrieval. Additionally, I would expect the modality to offer more flexibility for neural network development rather than requiring a specific encoder design. Furthermore, I would expect more groundbreaking improvements, but the enhancements in classification and retrieval are not significant.

**Questions:**

1 Is there any support or reference for the claim in lines 69-71? Or is it just more convenient for the work to enable unconstrained for the remaining layers?

2 Will changing the architecture of the SDF network improve or harm the performance of the tasks?

3 How much storage space does the work need to store the level-set parameters for the shape analysis tasks?

4 How much VRAM usage does the work need to construct the level-set parameters?

5 In Figure 3(b), some points from different categories are mixed together. Doesn't this suggest that the constructed dataset may contain potential noise, which could misguide the classification task?

---

### Official Review · Reviewer_XDHD · 2024-11-02

**Soundness:** 2
**Presentation:** 2
**Contribution:** 2
**Rating:** 5
**Confidence:** 3

**Summary:**

The paper proposes an approach for shape analysis that extends beyond traditional 3D data by employing a continuous, numerical representation of 3D shapes. The authors introduce a hypernetwork designed to transform input shapes by adjusting the level-set parameters of the 3D model in alignment with rotation and translation. This representation enables effective semantic shape analysis within the SO(3) space, offering an advantage over Euclidean-based transformations for continuous shapes. Experimental results demonstrate the method’s effectiveness, showing improvements over prior techniques in tasks such as shape classification and pose estimation.

**Strengths:**

- The paper covers a comprehensive landscape of prior works.
- Reproducibility - The code has been released, which will help in reproducing the results.

**Weaknesses:**

Clarity and Presentation: The paper could benefit from improved clarity in communicating its main ideas. For example, the rationale behind certain design choices is not well-articulated, making it challenging to understand the intuition behind the approach. Specifically, Section 3.2 & 4.1 would benefit from a revision to enhance readability, as it is difficult to get an intuitive understanding of what is going on in these sections. Below are few of the points which need improvements:

- In section 3.2, why a 2 stage approach for predicting the parameter decomposition is used?
- Why trainable parameters are initialized as zero in line 236?
- What is the intuition behind Equation (6)?
- Many things in section 4.1 seems hard-coded. For e.g. What is the intuition behind concatenating the weights and biases of the first SDF layer? What is the intuition behind using parameters of only layer 2 and 7 of the SDF network?

Because of these points I feel section 3.2 and 4.1 needs a major rewrite.

Results and Evaluation: The experimental results are presented on relatively simple datasets, such as ModelNet40 and ShapeNet. It would be valuable to evaluate the method on more complex, large-scale datasets like Objaverse to better assess its robustness and scalability. Will this method work in more modern datasets? Some quantitative or qualitative evaluation on these dataset is helpful. Given the relatively small improvements in quantitative comparison in Table 2 and 4, I have doubts about the robustness of this approach. And hence, a more extensive evaluation on complex datasets like Objaverse is required.

**Questions:**

Please refer to weakness.

---

### Official Review · Reviewer_YLH1 · 2024-11-03

**Soundness:** 2
**Presentation:** 1
**Contribution:** 2
**Rating:** 3
**Confidence:** 4

**Summary:**

This submission advocates to consider level-set parameters, namely, parameters of deep SDF networks, as a representation of 3D shapes, and proposes algorithms built upon the representation for tasks including 6D pose estimation, classification and retrieval, and registration. The main contribution is a tailored-for design for taking rotation/translation of shapes into account.

**Strengths:**

+ The idea of considering neural parameters as data representation is interesting.

**Weaknesses:**

- The motivation of this submission is vague. The experimental section demonstrate tasks mostly belonging to point cloud processing, the motivation of leveraging SDF formulation is not justified.

- The exposition in Sec. 3.1 is poor, there is no motivation (or at least reference to existing works) on the formulation of hyper-network. Why is matrix-form entry necessary? How is it related to better encoding of rigid transforms?

- Similarly, Sec. 4.1 threw a pile of architecture details on the readers without any reasoning. I found it hard to get why the network leads to semantic understanding of shapes.

- The running time is not fully reported. It is only mentioned in the end of Sec. 5 that it takes 50 seconds for registration, which is clearly slow.

**Questions:**

1. I did not understand the example in Line 40. Throughout the work, level-set parameters seem to be related to certain neural network, while here it is related to geometry, what is the point?

2. What does “pseudo-normal distribution” mean in Line 50, it is not clear from the context.

3. The fairness of experimental results in Tab. 4 is not clear. Algorithm 2 essentially uniformly samples the rotation space, and chooses the best candidates of initialization. Did the authors implement the same for the baselines?

4. In Tab. 3, the reported results for VN-DGCNN is way lower than the original paper [Deng et al. 2021]. Beyond that, why do PointNet and DGCNN, which are not pose equivariant, achieve such high score under SO(3)/SO(3)?

---

### Note · Authors · 2024-11-28

I have read and agree with the venue's withdrawal policy on behalf of myself and my co-authors.